# Synthesis and Stereostructure-Activity Relationship of Novel Pyrethroids Possessing Two Asymmetric Centers on a Cyclopropane Ring

**DOI:** 10.3390/molecules24061023

**Published:** 2019-03-14

**Authors:** Takashi Taniguchi, Yasuaki Taketomo, Mizuki Moriyama, Noritada Matsuo, Yoo Tanabe

**Affiliations:** Department of Chemistry, School of Science and Technology, Kwansei Gakuin University, 2-1 Gakuen, Sanda, Hyogo 669-1337, Japan; taniguchi.t.am@m.titech.ac.jp (T.T.); dzs31988@kwansei.ac.jp (Y.T.); ext32196@kwansei.ac.jp (M.M.); nor87168@cf6.so-net.ne.jp (N.M.)

**Keywords:** pyrethroid, structure‒activity relationship, asymmetric synthesis, cyclopropane formation, two asymmetric centers, common mosquito, chiral discrimination, hydroxylation cross-coupling reaction, fenvalerate, etofenprox

## Abstract

2-Methylcyclopropane pyrethroid insecticides bearing chiral cyanohydrin esters or chiral ethers and two asymmetric centers on the cyclopropane ring, were synthesized. These compounds were designed using a “reverse connection approach” between the isopropyl group in Fenvalerate, and between two dimethyl groups in an Etofenprox analogue (the methyl, ethyl form), respectively. These syntheses were achieved by accessible ring opening reactions of commercially available (±)-, (*R*)-, and (*S*)-propylene oxides using 4-chlorobenzyl cyanide anion as the crucial step, giving good overall yield of the product with >98% ee. The insecticidal activity against the common mosquito (*Culex pipiens pallens*) was assessed for pairs of achiral diastereomeric (1*R**,2*S**)-, (1*R**,2*R**)-cyanohydrin esters, and (1*R**,2*S**)-, (1*R**,2*R**)-ethers; only the (1*R**,2*R**)-ether was significantly effective. For the enantiomeric (1*S*,2*S*)-ether and (1*R*,2*R*)-ether, the activity was clearly centered on the (1*R*,2*R*)-ether. The present stereostructure‒activity relationship revealed that (i) cyanohydrin esters derived from fenvalerate were unexpectedly inactive, whereas ethers derived from etofenprox were active, and (ii) apparent chiral discrimination between the (1*S*,2*S*)-ether and the (1*R*,2*R*)-ether was observed. During the present synthetic study, we performed alternative convergent syntheses of Etofenprox and novel 4-EtO-type (1*S*,2*S*)- and (1*R*,2*R*)-pyrethroids from the corresponding parent 4-*Cl*-type pyrethroids, by utilizing a recently-developed hydroxylation cross-coupling reaction.

## 1. Introduction

Chiral discrimination between the diastereomers and enantiomers is a pivotal subject [1,2] concerning the development of synthetic pyrethroid insecticides [1,2,3,4,5,6]. The original natural chrysanthemic acid components possess cyclopropane structures with 1*R*,3*R* (or 3*S*) chiral centers as the insecticidally-active ingredient. Fenvalerate was the first distinctive non-cyclopropane-type synthetic pyrethroid to be reported, and the active *S*-enantiomer is commercially available as esfenvalerate (Figure 1) [7]. This concept on this discovery was interpreted by using the disconnection approach on the cyclopropane bond in natural chrysanthemic acids. The development of the ether-type pyrethroid etofenprox, on the other hand, overturned the prevailing belief that pyrethroids are composed of common ester-type moieties. In an independent investigation, the CSIRO (Commonwealth Scientific and Industrial Research Organisation, Australia) group disclosed pyrethroids containing a *gem*-dihalocyclopropane, cycloprothrin [8,9] the *S*-enantiomer of which is the active ingredient and is likely superimposable on esfenvalarate (Figure 2).

In our continuing synthetic studies of cyclopropane chemistry [10,11], we previously reported three synthetic designs and the structure activity relationships (SARs) of cyclopropane pyrethroids **I**, **II**, and **III** with three chiral centers (Figure 2) [12,13,14,15]. Apparent chiral discrimination among three sets of eight stereoisomers revealed that each enantiomer (1*R*,2*S*,3*S*) was an insecticidally-active form. Taking this background into account, we envisaged a further investigation of the synthesis of simpler and more accessible novel cyclopropane-type pyrethroids with two chiral centers: cyanohydrin ester-type **1** and ether-type **2** (Scheme 1). The design of **1** and **2** involved a “reverse connection approach” between the **a** and **b** positions in fenvalerate and between the **c** and **d** positions in an etofenprox analogue (methyl and ethyl form). Furthermore, the present work was performed using an accessible chiral synthetic method, eliminating the tedious optical resolution and elaborated asymmetric cyclopropanation procedures used for **I**‒**III**. Convergent synthesis of etofenprox and the present 4-EtO-type pyrethroids from the parent 4-*Cl*-type pyrethroids utilizing a recently developed hydroxylation cross-coupling reaction is also described.

## 2. Results and Discussion

### 2.1. Syntheses of Novel Pyrethroids Possessing two Asymmetric Centers on the Cyclopropane Ring

Synthetic routes for key cyclopropane precursors **4** are outlined in Scheme 2. Ring-opening reactions of commercially available (±)-, (*R*)-, and (*S*)-propylene oxides by 4-chlorobenzyl cyanide anion give the corresponding secondary alcohol intermediates. Tosylation of the intermediates produces 2-(4-chlorophenyl)-4-tosyloxynitriles (±)-**3**, (4*R*)-**3**, and (4*S*)-**3**. An S_N_2-type ring-closing reaction using mixtures of (4*R*)-**3** and (4*S*)-**3** produces the key cyclopropane nitriles (1*R**,2*S**)-**4** and (1*R**,2*R**)-**4** as diastereomers. In a similar manner, two sets of nitriles (1*R*,2*S*)-**4**/(1*S*,2*S*)-**4** and (1*S*,2*R*)-**4**/(1*R*,2*R*)-**4** can be prepared from (4*R*)- and (4*S*)-**3**, respectively.

The successful accessible cyclopropane formation steps are shown in Scheme 3. The LHMDS-mediated ring-opening reaction of 4-chlorobenzyl cyanide with (±)-propylene oxide gave intermediary alcohol (±)-**5** with a yield of 70% with ca. 6:4 diastereoselectivity according to the reported method [16]. Alcohol (±)-**5** was converted to tosylate (±)-**2** with excellent yield using TsCl/Et_3_N/cat.Me_3_N•HCl reagent [17] (96%, dr = 65:35). We then investigated the crucial cyclopropane ring-closing reaction. Hiyama and Takehara’s group reported a relevant study on stereoselective reactions of 4-methoxybenzyl cyanide with chiral epoxyoctane using three different bases (NaH, *t*BuOK, and LDA) [16]. To intentionally obtain diastereomers (1*R**,2*S**)-**4** and (1*R**,2*R**)-**4** in equal amounts, we applied non-stereoselective conditions using *t*BuOLi, as shown in Table 1. The relative stereochemical assignment of cyclopropanes **4** was determined according to the reported method [16]. Based on the criterion for the deshielding effect of a *cis*-located benzene ring, methyl protons (0.83 ppm) in (1*R**,2*R**)-**4** appeared at a higher chemical shift than those (1.47 ppm) in (1*R**,2*S**)-**4**.

Hydrolysis and successive reduction of (1*R**,2*S**)-**4** and (1*R**,2*R**)-**4** were next examined (Scheme 4). Conventional hydrolysis of (1*R**,2*R**)-**4** using NaOH/MeOH‒H_2_O gave carboxylic acid (1*R**,2*R**)-**6** with a yield of 84%, whereas (1*R**,2*S**)-**4** resisted these conditions and was only produced with a yield of 30‒35%, probably due to the higher *cis*-methyl group stereocongestion. To address this issue, DIBAL reduction and a Pinnick oxidation reaction sequence were applied to (1*R**,2*S**)-**4** to successfully produce acid (1*R**,2*S**)-**6** with an overall yield of 76%. LAH reduction of carboxylic acids (1*R**,2*S**)-**6** and (1*R**,2*R**)-**6** gave alcohol intermediates (1*R**,2*S**)-**7** and (1*R**,2*R**)-**7** with yields of 92% and 94%, respectively.

Based on these results, cyanohydrin ester formation using acids (1*R**,2*S**)-**6** and (1*R**,2*R**)-**6** was carried out. Conventional condensation reactions of (1*R**,2*S**)-**6** and (1*R**,2*R**)-**6** with bromo(3-phenoxyphenyl)acetonitrile [9] produced target cyanohydrin esters (1*R**,2*S**)-**8** and (1*R**,2*R**)-**8** in yields of 73% and 79%, respectively (Scheme 5). As addressed in the bioassay section (vide infra), neither cyanohydrin ester (1*R**,2*S**)-**8** nor (1*R**,2*R**)-**8** exhibited significant insecticidal activity. Accordingly, we abandoned further synthesis of the corresponding chiral compounds.

We next turned our attention to the synthesis of all stereoisomers of achiral and chiral ether-type pyrethroids **9** (Scheme 6). Conventional ether formation of alcohols (1*R**,2*S**)-**7** and (1*R**,2*R**)-**7** with 3-phenoxybenzyl bromide produced a pair of achiral ethers, (1*R**,2*S**)-**9** and (1*R**,2*R**)-**9**, with good yield. Based on the pilot approach, all four sets of chiral ethers, (1*R*,2*S*)-**9**, (1*S*,2*S*)-**9**, (1*S*,2*R*)-**9**, and (1*R*,2*R*)-**9**, were successfully synthesized. It is noteworthy that clean S_N_2 reactions using tosylates (4*R*)-**3** and (4*S*)-**3** proceeded smoothly to produce four key cyclopropane carbonitrile nitrile precursors, (1*R*,2*S*)-**4**, (1*S*,2*S*)-**4**, (1*S*,2*R*)-**4**, and (1*R*,2*R*)-**4**, with excellent optical purities (all 98% ee, as determined by HPLC analysis). Related chirality-induced cyclopropane-forming reactions have been addressed in previous reports, namely a representative review [18], syntheses of liquid crystals [16], and syntheses of dihydroxy vitamin D_3_ analogues [19], petrosterol [20], chiral cyclobutanes [21], bicifadin [22], norchrysanthemic acid [23], and grandisol [23].

Because the bioassay section indicated that only the achiral ether (1*R**,2*R**)-**9** exhibited significant insecticidal activity (vide infra), chiral ethers (1*S*,2*S*)-**9** and (1*R*,2*R*)-**9** were synthesized to evaluate chiral discrimination.

To obtain EtO-type parent etofenprox analogues, we designed convergent syntheses of chiral ethers (1*S*,2*S*)-**10** and (1*R*,2*R*)-**10** from (1*S*,2*S*)-**9** and (1*R*,2*R*)-**9**, respectively (Scheme 7). Fortuitously, a hydroxylation cross-coupling procedure [24] developed by Buchwald’s group fulfilled this purpose; the reaction using KOH/Pd(dba)_2_/*t*Bu-XPhos catalysis and successive ethylation furnished the desired ethers (1*S*,2*S*)-**10** and (1*R*,2*R*)-**10** with a good overall yield. Despite the relatively large molecules (Molecular wight = 346.9), hydroxylation using **13** proceeded smoothly with a 94% yield without loss of the cyclopropane and ether functional groups.

### 2.2. Alternative Synthesis of Achiral Etofenprox Analogues for Reference Compounds

Two reported cyclopropane-type etofenprox analogues, **I** (**13**) [25] and **II** (**14**) [25], the parent compounds of **9**, were synthesized as reasonable reference compounds for comparing the insecticidal activity (Scheme 8). The KOH/cat. tetrabutylammonium bromide (TBAB)-mediated reaction of 4-chlorobenzyl cyanide with 1,2-dibromoethane gave cyclopropane carbonitrile **11** with an 80% yield. The aforementioned conventional stepwise reactions starting from **11** led to the production of etofenprox analogue-**I** (**13**) through alcohol **12** with an overall yield of 68%. Hydroxylation cross-coupling using analogue **I** (**13**) also proceeded smoothly in this case to produce analogue **II** (**14**) with an 80% yield (2 steps).

This successful outcome prompted us to apply the conversion process in Scheme 8 to the concise synthesis of etofenprox itself, starting from readily available methyl 4-chlorophenylacetate (**15**) (Scheme 9).

Dimethylation of **15** using 2MeI/2NaH reagent, followed by LAH-reduction, gave chloro-type precursor **17** with an 85% yield (2 steps). Conventional etherification of **17** with 3-phenoxybenzyl bromide afforded precursor **18** (99%). Following the aforementioned hydroxylation and ethylation sequence, etofenprox was produced with a 72% yield (two steps). The reported method utilized *meta*-Cl-surrogate for Friedel‒Crafts alkylation and dechlorination [26]. The present method represents an alternative synthetic method for etofenprox.

### 2.3. Insecticidal Activity and Stereostructure‒Activity Relationship

Insecticidal activity against the common mosquito (*Culex pipiens pallens*) was assessed for pairs of achiral diastereomeric cyanohydrin esters (1*R**,2*S**)-**8**, (1*R**,2*R**)-**8**, and ethers (1*R**,2*S**)-**9**, (1*R**,2*R**)-**9** using fenvalerate as a reference compound (Table 2). The salient results are as follows: (i) neither cyanohydrin esters (1*R**,2*S**)-**8** nor (1*R**,2*R**)-**8** exhibited significant insecticidal activity, even at 500 ppm. This result did not support our initial hypothesis, described in the introduction section, i.e., imaginary cyclopropane formation of fenvalerate (“reverse connection approach”) decreased the activity. Compared with fenvalerate, the solid cyclopropane conformation in **8** may inhibit flexible-induced fitting for a conceivable “pyrethroid receptor”. (ii) In contrast, ethers (1*R**,2*S**)-**9** and (1*R**,2*R**)-**9** exhibited significant activities at 500 ppm, and (1*R**,2*R**)-**9** was apparently more active than (1*R**,2*S**)-**9** at 250 ppm. With these promising results in our hand, we focused our attention on chiral discrimination between enantiomeric ether compounds (1*R*,2*R*)-**9** and (1*S*,2*S*)-**9**.

Table 3 lists the results of insecticidal activities of pairs of chiral ethers (1*S*,2*S*)-**9**, (1*R*,2*R*)-**9**, and EtO-analogues (1*S*,2*S*)-**10**, (1*R*,2*R*)-**10**, in which achiral ethers **13**, **14**, and **18**, and etofenprox are selected as reference compounds. The salient features are as follows; (i) Regarding Cl-analogues, (1*R*,2*R*)-**9** exhibited higher activity than (1*S*,2*S*)-**9** at 250 ppm; (ii) both (1*S*,2*S*)-**9** and (1*R*,2*R*)-**9** were inactive at 50 ppm; (iii) EtO-analogues (1*R*,2*R*)-**10** exhibited a significant activity compared with (1*S*,2*S*)-**10** at 50 ppm – this clear chiral discrimination between the enantiomers (1*S*,2*S*)-**9** or **-10** and (1*R*,2*R*)-**9** or -**10** is noteworthy; (iv) both the reported isosteric cyclopropane compounds **13** and **14** [25] ande exhibited nearly equal activities; (v) All the EtO-analogues were stronger than the *Cl*-analogues.

To summarize the results, all of the presented 2-methylcyclopropane pyrethroids with two asymmetric centers exhibited clear chiral discrimination between the enantiomers, although the insecticidal activity was slightly weaker than that of the reported parent reported achiral cyclopropane and dimethyl pyrethroids. Notably, in contrast to our expectations based on a couple of *three* asymmetric-center cyclopropane pyrethroids previously reported by our group [13,14], the ether-type, not the cyanohydrin ester type, was the active form.

## 3. Materials and Methods

All reactions were carried out in oven-dried glassware under an argon atmosphere. Flash column chromatography was performed with silica gel 60 (230–400 mesh ASTM, Merck, Darmstat, Germany). TLC analysis was performed on Merck 0.25 mm Silicagel 60 F_254_ plates. Melting points were determined on a hot stage microscope apparatus (ATM-01, AS ONE, Osaka, Japan) and were uncorrected. NMR spectra were recorded on a JEOLRESONANCE EXC-400 or ECX-500 spectrometer (JEOL, Akishima, Japan) operating at 400 MHz or 500 MHz for ^1^H-NMR, and 100 MHz and 125 MHz for ^13^C-NMR. Chemical shifts (δ ppm) in CDCl_3_ were reported downfield from TMS (= 0) for ^1^H-NMR. For ^13^C-NMR, chemical shifts were reported in the scale relative to CDCl_3_ (77.00 ppm) as an internal reference. IR Spectra were recorded on FT/IR-5300 spectrophotometer (JASCO, Akishima, Japan). Mass spectra were measured on a JMS-T100LC spectrometer (JEOL). HPLC data were obtained on a SHIMADZU (Kyoto, Japan) HPLC system (consisting of the following: LC-20AT, CMB20A, CTO-20AC, and detector SPD-20A measured at 254 nm) using Chiracel AD-H or Ad-3 column (Daicel, Himeji, Japan, 25 cm) at 25 °C. Optical rotations were measured on a JASCO DIP-370 (Na lamp, 589 nm). The synthetic procedure of **11**, **12**, **13**, **14**, **16**, **17**, **18**, and etofenprox are available in the Appendix A. ^1^H-, ^13^C-NMR spectra for compounds (±)-**5**, (±)-**3**, (*1R**,*2S**)-**4**, (*1R**,2R*)-**4**, (*1R**,*2S**)-**6**, (*1R**,*2R**)-**6**, (*1R**,*2S**)-**8**, (*1R**,*2R**)-**8**, (*1R**,*2S**)-**7**, (*1R**,*2R**)-**7**, (*1R**,*2S**)-**9**, (*1R**,*2R**)-**9**, (*1R*,*2R*)-**10** can see in the Appendix A.

*(±)-2-(4-Chlorophenyl)-4-hydroxypentanenitrile*: (±)-**5** (diastereo mixtures)



LHMDS (1.1 M in THF, 10 mL, 11 mmol) was added to a stirred suspension of 4-chlorobenzyl cyanide (1.52 g, 10 mmol) in THF (13 mL) at 0–5 °C under an Ar atmosphere, and the mixture was stirred at that temperature for 1 h. (±)-Propylene oxide (2.1 mL, 30 mmol) was added to the mixture, which was stirred for 1 h. The mixture was quenched with 1M-HCl aq., which was extracted three times with AcOEt. The combined organic phase was washed with brine, dried (Na_2_SO_4_) and concentrated. The obtained crude oil was purified by SiO_2_-gel column chromatography (hexane/AcOEt = 10:1 to 1:2) to give the desired product (±)-**3** (1.48 g, 70%, dr = 65:35). Pale yellow oil; ^1^H-NMR (400 MHz, CDCl_3_) δ = 1.23 (d, *J* = 6.4 Hz, 3H × 7/20), 1.29 (d, *J* = 6.0 Hz, 3H × 13/20), 1.50–1.70 (brs, 1H), 1.79–1.86 (m, 1H × 13/20), 1.89–2.01 (m, 1H), 2.10–2.17 (m, 1H × 7/20), 3.56–3.67 (m, 1H × 13/20), 4.04 (dd, *J* = 5.5 Hz, *J* = 9.6 Hz, 1H × 7/20), 4.14–4.21 (m, 1H), 7.28–7.38 (m, 4H); ^13^C-NMR (125 MHz; CDCl_3_) δ = 23.6, 23.8, 32.7, 33.7, 43.8, 44.7, 63.9, 65.0, 120.4, 121.1, 128.4, 129.0, 129.2 (2C), 133.7, 133.8, 134.0, 134.4; ν_max_ (neat)/cm^−1^ 3424, 2968, 2241, 1493, 1410, 1375, 1130, 1093, 1014, 953, 932, 845, 822, 791, 779, 718; HRMS (DART) calcd for C_11_H_12_ClNO [M + H]^+^ 210.0686, found 210.0688.

*(4R)-2-(4-Chlorophenyl)-4-hydroxypentanenitrile*: (*R*)-**5** (diastereo mixtures)



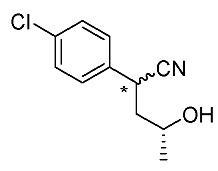



Following the similar procedure for the preparation of (±)-**3**, the reaction of chlorobenzyl cyanide (758 mg, 5.0 mmol), (*R*)-propylene oxide (1.05 mL, 15 mmol), and LHMDS (1.1 M in THF, 5.0 mL, 5.5 mL), and the successive SiO_2_-gel column chromatographic purification gave the desired product (*R*)-**5** (813 mg, 78%, dr = 65:35). Pale yellow oil.

*(4S)-2-(4-Chlorophenyl)-4-hydroxypentanenitrile*: (*S*)-**5** (diastereo mixtures)



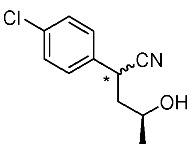



Following the similar procedure for the preparation of (±)-**3**, the reaction of chlorobenzyl cyanide (1.52 g, 10 mmol), (*S*)-propylene oxide (2.1 mL, 30 mmol) and LHMDS (1.1 M in THF, 8.46 mL, 1.1 mL), and the successive SiO_2_-gel column chromatographic purification gave the desired product (*S*)-**5** (1.57 g, 75%, dr = 65:35). Pale yellow oil.

*(±)-4-(4-Chlorophenyl)-4-cyanobutan-2-yl 4-methylbenzenesulfonate*: (±)-**3** (diastereo mixtures)



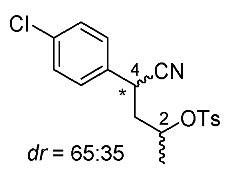



TsC1 (286 mg, 1.5 mmol) in MeCN (1.0 mL) was added to a stirred solution of (±)-**3** (210 mg, 1.0 mmol, dr = 65:35), Et_3_N (202 mg, 2.0 mmol), and Me_3_N·HC1 (10 mg, 0.1 mmol) in MeCN (1.0 mL) at 0–5 °C, and the mixture was stirred for 1 h. Water was added to the mixture, which was extracted twice with EtOAc. The combined organic phase was washed with water and brine, dried (Na_2_SO_4_) and concentrated. The obtained crude oil was purified by SiO_2_-gel column chromatography (hexane/AcOEt = 30:1 to 10:1) to give the desired product (±)-**3** (350 mg, 96%, dr = 65:35). Pale yellow oil; ^1^H-NMR (400 MHz, CDCl_3_) δ = 1.27 (d, *J* = 6.4 Hz, 3H × 7/20), 1.32 (d, *J* = 6.4 Hz, 3H × 13/20), 1.99–2.11 (m, 2H,), 2.47 (s, 3H), 3.81 (dd, *J* = 5.5 Hz, *J* = 10.1 Hz, 1H × 13/20), 3.90 (dd, *J* = 5.5 Hz, *J* = 8.7 Hz, 1H × 7/20), 4.51–4.59 (m, 1H × 7/20), 4.77–4.85 (m, 1H × 13/20), 7.20–7.40 (m, 6H), 7.77–7.79 (m, 2H × 7/20), 7.85–7.87 (m, 2H × 13/20); ^13^C-NMR (125 MHz; CDCl_3_) δ = 20.3, 20.6, 21.3 (2C), 31.9, 33.0, 41.4, 42.3, 75.7, 76.6, 119.1, 119.8, 127.3, 127.5, 128.2, 128.8, 129.0, 129.1, 129.7, 129.8, 132.8, 133.1, 133.2, 133.4, 133.8, 134.0, 144.8, 145.0.; ν_max_ (neat)/cm^−1^ 3022, 2984, 2243, 1597, 1493, 1362, 1188, 1175, 1096, 1059, 1016, 924, 891, 816, 750, 687, 662.

*(2R)-4-(4-Chlorophenyl)-4-cyanobutan-2-yl 4-methylbenzenesulfonate*: (*R*)-**3** (diastereo mixtures)



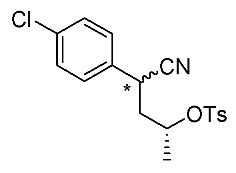



Following the similar procedure for the preparation of (±)-**3**, the reaction of (*R*)-**5** (629 mg, 3.0 mmol, dr = 50:50), TsCl (858 mg, 4.5 mmol), Et_3_N (607 mg, 6.0 mmol), and Me_3_N·HCl (29 mg, 0.3 mmol), and the successive SiO_2_-gel column chromatographic purification gave the desired product (*R*)-**3** (980 mg, 90%, dr = 65:35). Pale yellow oil.

*(2S)-4-(4-Chlorophenyl)-4-cyanobutan-2-yl 4-methylbenzenesulfonate*: (*S*)-**3** (diastereo mixtures)



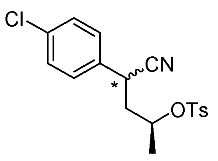



Following the similar procedure for the preparation of (±)-**3**, the reaction of (*S*)-**5** (1.54 g, 7.34 mmol, dr = 50:50), TsCl (2.10 g, 11.0 mmol), Et_3_N (1.49 g, 14.7 mmol), and Me_3_N·HCl (70 mg, 0.7 mmol), and the successive SiO_2_-gel column chromatographic purification gave the desired product (*S*)-**3** (2.45 g, 92%, dr = 65:35). Pale yellow oil.

*(1R*,2S*)- and (1R*,2R*)-1-(4-Chlorophenyl)-2-methylcyclopropane-1-carbonitrile*: (1*R**,2*S**)-**4** and (1*R**,2*R**)-**4**



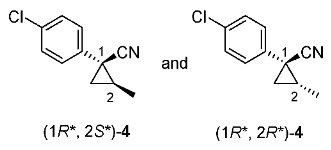



Tosylate (±)-**3** (65:35: diastereo mixtures) (942 mg, 2.59 mmol) in Et_2_O (13 mL) was added to a stirred suspension of *t*BuOLi (249 mg, 3.11 mmol) in Et_2_O (13 mL) at −78 °C under Ar atmosphere, followed by being stirred at the same temperature for 0.5 h. The mixture was allowed to warm to 0–5 °C, and was stirred for 2 h. The mixture was quenched with water, which was extracted three times with AcOEt. The combined organic phase was washed with brine, dried (Na_2_SO_4_) and concentrated. The obtained crude oil was purified by SiO_2_-gel column chromatography (hexane/AcOEt = 50:1) to give (1*R*,*2*S**)-**4** (175mg, 35%) and (1*R*,*2*R**)-**4** (161 mg, 32%).

(1*R*,*2*S**)-**4:** Colorless oil; ^1^H-NMR (400 MHz, CDCl_3_) δ = 1.41 (dd, *J* = 10.1 Hz, *J* = 11.9 Hz, 1H), 1.47–1.49 (m, 3H), 1.56–1.60 (m, 2H), 7.18–7.21 (m, 2H), 7.29–7.33 (m, 2H); ^13^C-NMR (125 MHz, CDCl_3_): δ 15.9, 20.0, 25.3(2C), 120.4, 126.8, 128.8, 133.1, 135.2.; ν_max_ (neat)/cm^−1^ 2968, 2934, 2234, 1495, 1445, 1402, 1369, 1121, 1096, 1015, 889, 849, 827, 764, 727.; HRMS (DART) calcd for C_11_H_10_OClN [M + H]^+^ 192.0580, found 192.0573.

(1*R*,*2*R**)-**4:** Colorless oil; ^1^H-NMR (400 MHz, CDCl_3_) δ = 0.83 (d, *J* = 6.5 Hz, 3H), 1.26 (dd, *J* = 5.5 Hz, *J* = 6.9 Hz, 1H), 1.74 (dd, *J* = 5.5 Hz, *J* = 9.3 Hz, 1H), 1.83–1.94 (m, 1H), 7.25–7.38 (m, 4H); ^13^C-NMR (125 MHz; CDCl_3_): δ 13.6, 17.4, 19.9, 23.2, 123.2, 128.9, 130.7, 131.0, 134.1.; ν_max_ (neat)/cm^−1^ 2966, 2230, 1493, 1445, 1396, 1366, 1315, 1124, 1094, 1043, 1014, 997, 908, 862, 827, 789, 729.

*(1R,2S)- and (1S,2S)-1-(4-Chlorophenyl)-2-methylcyclopropane-1-carbonitrile*: (1*R*,2*S*)-**4** and (1*S*,2*S*)-**4**



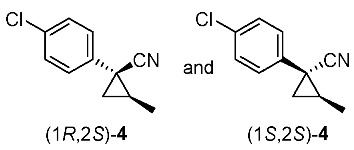



Following the similar cyclization procedure described abave, (1*R,*2*S*)-**4** (45%) and (1*S*,2*S*)-**4** (39%) were prepared from (2*R*)-**3**.

(1*R*,2*S*)**-4**: Colorless oil; [α]${}_{D}^{21}$ +64.9 (*c* 1.00, CHCl_3_); 98% ee, HPLC analysis (Daicel Chiralcel OD-H column, solvent: hexane/ethanol = 150:1, flow rate 0.80 mL/min, 254 nm UV detector), t_R_(racemate) = 10.38 min and 11.18 min. t_R_[(1*R,2S*)-form] = 9.74 min

(1*S*,2*S*)-**4**: White colored solid; m.p. 73–75 °C, [α]${}_{D}^{17}$ +36.6 (*c* 0.7, CHCl_3_); >98% ee, HPLC analysis (Daicel Chiralcel OD-H column, solvent: hexane/ethanol = 300:1, flow rate 0.80 mL/min, 254 nm UV detector), t_R_(racemate) = 17.41 min and 18.55 min. t_R_[(1*S*,2*S*)-form] = 19.01 min.

*(1S,2R) and (1R,2R)-1-(4-Chlorophenyl)-2-methylcyclopropane-1-carbonitrile*: (1*S*,2*R*)-**4**, (1*R*,2*R*)-**4**



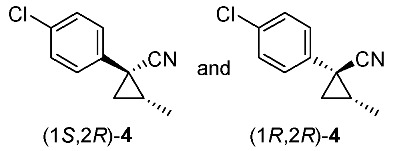



Following the similar cyclization procedure described above, (1*S,*2*R*)-**4** (40%) and (1*R*,2*R*)-**4** (38%) were prepared from (2*S*)-**3**.

(1*S*,2*R*)-**3**: Colorless oil; [α]${}_{D}^{23}$ −62.6 (*c* 0.25, CHCl_3_); 98% ee, HPLC analysis (Daicel Chiralcel OD-H column, solvent: hexane/ethanol = 150:1, flow rate 0.80 mL/min, 254 nm UV detector), t_R_(racemate) = 10.38 min and 11.18 min. t_R_[(1*S,2R*)-form] = 11.36 min.

(1*R*,2*R*)-**4**: White colored solid; m.p. 73-74 °C; [α]${}_{D}^{20}$ −33.4 (*c* 0.33, CHCl_3_); >98% ee, HPLC analysis (Daicel Chiralcel OD-H column, solvent: hexane/ethanol = 300:1, flow rate 0.80 mL/min, 254 nm UV detector), t_R_(racemate) = 17.41 min and 18.55 min. t_R_[(1*R,*2*R*)-form] = 15.01 min.

*(1R*,2S*)-1-(4-chlorophenyl)-2-methylcyclopropane-1-carboxylic acid*: (1*R**,2*S**)-**6**



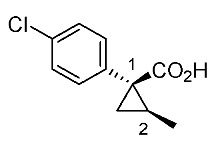



DIBAL (1.01 M in toluene, 1.98 mL, 2.0 mmol) was added slowly to a stirred solution of (1*R**,2*S**)-**4** (192 mg, 1.0 mmol) in toluene (10 mL) at −78 °C under an Ar atmosphere. The mixture was allowed to warm to 20–25 °C over 1 h, and then the mixture was quenched with MeOH. Then, potassium sodium tartrate tetrahydrate solution was added to the mixture. After stirring for 0.5 h, the mixture was extracted three times with EtOAc. The combined organic phase was washed with brine, dried (Na_2_SO_4_) and concentrated. The obtained crude aldehyde product (140 mg) was used for the next step without purification. To a stirred solution of the obtained aldehyde (140 mg), 2-methyl-2-butene (561 mg, 8.0 mmol) in *t*BuOH (9 mL), water (4 mL) were added NaH_2_PO_4_ (480 mg, 4.0 mmol) in water (4 mL) and NaClO_2_ (217 mg, 2.4 mmol) in water (4 mL) at 20–25 °C, and the mixture was stirred at that temperature for 1 h. The reaction mixture was quenched with water and extracted with AcOEt three times. The combined organic phase was washed with brine, dried (Na_2_SO_4_) and concentrated. Sat. NaHCO_3_ aq. was added to the residue, which was wasehd with Et_2_O two times. The separated aqueous phase was acidified with 1M HCl aq. solution, which was extracted with AcOEt three times. The combined organic phase was washed with brine, dried (Na_2_SO_4_) and concentrated to give the desired product (1*R**,2*S**)**-6** (160 mg, 76% for two steps). White colored solid; m.p. 136–137 °C; ^1^H-NMR (400 MHz, CDCl_3_) δ = 1.33 (dd, *J* = 4.5 Hz, 8.9 Hz, 1H), 1.37 (d, *J* = 6.2 Hz, 3H), 1.56 (dd, *J* = 4.5 Hz, 7.6 Hz, 1H), 1.59–1.69 (m, 1H), 7.26–7.27 (m, 4H); ^13^C-NMR (125 MHz; CDCl_3_): δ 12.7, 22.6, 25.3, 33.7, 128.3, 131.6, 133.0, 139.0, 178.6.; ν_max_ (neat)/cm^−1^ 3011, 2880, 2598, 2363, 1686, 1495, 1420, 1312, 1213, 1092, 1013, 889, 829, 754, 719.

*(1R,2S)-1-(4-Chlorophenyl)-2-methylcyclopropane-1-carboxylic acid*: (1*R*,2*S*)-**6**



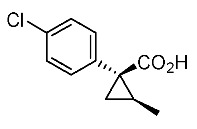



Following the similar DIDAL-reduction and Pinnick oxidation procedure described above, (1*R,2S*)-**6** was prepared from (1*R,2S*)-**4** (80%, two steps). White colored crystal; m.p. 104–106 °C; [α]${}_{D}^{16}$ +120.6 (*c* 0.16, CHCl_3_).

*(1S,2R)-1-(4-Chlorophenyl)-2-methylcyclopropane-1-carboxylic acid*: (1*S*,2*R*)-**6**



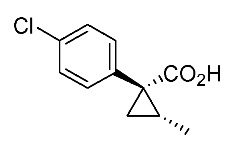



Following the similar DIDAL-reduction and Pinnick oxidation procedure described above, (1*S,2R*)-**6** (79%, 2 steps) was prepared from (1*S,2R*)-**4** (79%, two steps). White colored crystals; m.p. 102–105 °C; [α]${}_{D}^{17}$ −116.4 (*c* 0.97, CHCl_3_).

*(1R*,2R*)-1-(4-Chlorophenyl)-2-methylcyclopropane-1-carboxylic acid*: (1*R**,2*R**)-**6**



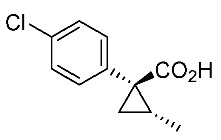



A mixture of (1*R**,2*R**)-**4** (381 mg, 2.0 mmol) and NaOH (1.6 g, 40 mmol) in MeOH (10.0 mL) and H_2_O (6.6 mL) was heated at 100 °C for 16 h. After cooling down, the mixture was quenched with water, which was washed with Et_2_O two times. The separated aqueous phase was acidified with 1M HCl aq. solution, which was extracted with EtOAc three times. The combined organic phase was washed with brine, dried (Na_2_SO_4_) and concentrated to give the desired product (1*R**,2*R**)-**6** (358 mg, 84%). White colored crystals: m.p. 130–133 °C; ^1^H-NMR (400 MHz, CDCl_3_) δ = 0.83 (d, *J* = 6.4 Hz, 3H), 1.09 (dd, *J* = 4.6 Hz, *J* = 6.9 Hz, 1H), 1.80 (dd, *J* = 4.6 Hz, *J* = 9.2 Hz, 1H), 1.90–1.98 (m, 1H), 7.18–7.22 (m, 2H), 7.28–7.32 (m, 2H); ^13^C-NMR (125 MHz; CDCl_3_): δ 12.7, 22.6, 25.3, 33.7, 128.3, 131.6, 133.0, 139.0, 178.6.; ν_max_ (neat)/cm^−1^ 3011, 2880, 2598, 2363, 1686, 1495, 1420, 1312, 1213, 1092, 1013, 943, 889, 829, 754, 719; HRMS (DART) calcd for C_11_H_11_ClO_2_ [M + H]^+^ 211.0526, found 211.0523.

*(1R,2R)-1-(4-Chlorophenyl)-2-methylcyclopropane-1-carboxylic acid*: (1*R*,2*R*)-**6**



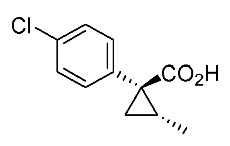



Following the similar hydroxylation procedure described above, (1*S,2S*)-**6** was prepared from (1*S,2S*)-**4** (92%). White colored crystals; m.p. 103–104 °C; [α]${}_{D}^{18}$ −19.0 (*c* 1.06, CHCl_3_)

*(1S,2S)-1-(4-Chlorophenyl)-2-methylcyclopropane-1-carboxylic acid*: (1*S*,2*S*)-**6**



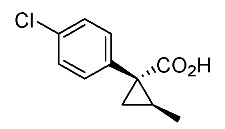



Following the similar hydroxylation procedure described above, (1*S,2S*)-**6** was prepared from (1*S,2S*)-**4** (92%). White colored crystals; m.p. 101–104 °C; [α]${}_{D}^{20}$ +17.8 (*c* 1.00, CHCl_3_).

*Cyano(3-phenoxyphenyl)methyl(1R*,2S*)-1-(4-chlorophenyl)-2-methylcyclopropane-1-carboxylate*: (1*R**,2*S**)-**8**



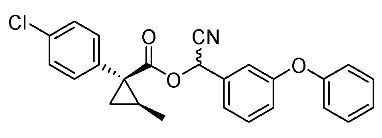



Et_3_N (0.06 mL, 0.40 mmol) was added to a stirred mixture of (1*R**,2*S**)-**6** (77 mg, 0.36 mmol) and bromo(3-phenoxyphenyl)acetonitrile (104 mg, 0.36 mmol) in acetone (0.72 mL) at 0–5 °C, and the mixture was stirred at 20–25 °C for 2 h. 1M HCl aq. solution was added to the mixture, which was extracted twice with ether. The combined organic phase was washed with water, brine, dried (Na_2_SO_4_) and concentrated. The obtained crude oil was purified by SiO_2_-gel columm chromatography (hexane/EtOAc = 30:1) to give the desired product (1*R**,2*S**)-**8** (110 mg, 73%). Yellow colored oil; ^1^H-NMR (400 MHz, CDCl_3_) δ = 1.26 (d, *J* = 6.0 Hz, 3H × 1/2), 1.32 (d, *J* = 6.0 Hz, 1H × 1/2), 1.32 (dd, *J* = 4.1 Hz, 4.6 Hz, 1H × 1/2), 1.36 (dd, *J* = 4.6 Hz, 8.7 Hz, 1H × 1/2), 1.58 (dd, *J* = 4.6 Hz, 5.0 Hz, 1H × 1/2), 1.60 (dd, *J* = 4.6 Hz, 5.0 Hz, 1H × 1/2), 1.63–1.73 (m, 1H), 6.31 (s, 1H × 1/2), 6.34 (s, 1H × 1/2), 6.87–7.49 (m, 13H); ^13^C-NMR (125 MHz) δ = 12.8, 12.9, 22.5, 22.7, 25.0, 25.2, 33.8, 33.9, 62.6, 62.7, 115.7, 116.0, 116.9, 117.0, 119.2, 119.4, 120.1, 120.2, 121.4, 121.6, 124.0, 124.1, 128.5 (2C), 123.0 (2C), 130.5, 130.6, 131.4 (2C), 133.3 (2C), 133.4, 133.6, 138.0 (2C), 156.2, 156.3, 158.0, 158.2, 170.2 (2C); ν_max_ (neat)/cm^−1^ 2931, 2360, 1734, 1585, 1487, 1155, 1244, 1090, 1015, 692. HRMS (DART) calcd for C_25_H_20_ClNO_3_ [M + H]^+^ 418.1210, found 418.1192.

*Cyano(3-phenoxyphenyl)methyl (1R*,2R*)-1-(4-chlorophenyl)-2-methylcyclopropane-1-carboxylate*: (1*R**,2*R**)-**8**



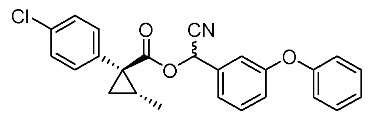



Following the similar esterification procedure described above, (1*R**,2*R**)-**8** was prepared from (1*R**,2*R**)-**6** (79%). Yellow colored oil; ^1^H-NMR (400 MHz, CDCl_3_) δ = 0.83 (d, *J* = 5.8 Hz, 3H × 1/2), 0.84 (d, *J* = 4.8 Hz, 3H × 1/2), 1.13 (dd, *J* = 4.5 Hz, 4.8 Hz, 1H × 1/2), 1.16 (dd, *J* = 4.1 Hz, 4.8 Hz, 1H × 1/2), 1.81 (dd, *J* = 4.5 Hz, 9.3 Hz, 1H × 1/2), 1.84 (dd, *J* = 4.5 Hz, 9.3 Hz, 1H × 1/2), 1.89–2.00 (m, 1H), 6.29 (s, 1H × 1/2), 6.32 (s, 1H × 1/2), 6.92–7.49 (m, 13H); ^13^C-NMR (125 MHz) δ = 15.2, 15.2, 23.6(2C), 24.1, 24.3, 32.7(2C), 62.6(2C), 115.8, 115.9, 116.8, 116.9, 119.3(2C), 119.3(2C), 120.0, 120.1, 121.4, 121.5, 124.0(2C), 128.4(2C), 129.9(2C), 130.5(2C), 132.7(2C), 133.1(2C), 133.4, 133.4, 133.5(2C), 156.2, 156.2, 158.1(2C), 172.2, 172.2; ν_max_ (neat)/cm^−1^ 2960, 2359, 1738, 1585, 1487, 1207, 1163, 1112, 1074, 1045, 745, 690.

*(1R*,2S*)-1-(4-Chlorophenyl)-2-methylcyclopropane-1-methanol*: (1*R**,2*S**)-**7**



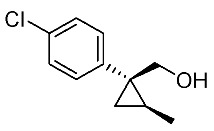



Acid (1*R**,2*R**)-**7** (126 mg, 0.6 mmol) in THF (3.0 mL) was added to a stirred suspension of LiAlH_4_ (36 mg) in THF (1.0 mL) at 0–5 °C under an Ar atmosphere, and the mixture was stirred at 20–25 °C for 2 h. Sat. NH_4_Cl aqueous solution was added to the mixture, which was filtered through Celite^®^ using EtOAc. The separated organic phase was washed with water, brine, dried (Na_2_SO_4_), and concentrated. The obtained crude oil was purified by SiO_2_-gel column chromatography (hexane/EtOAc = 10:1) to give the desired product (1*R**,2*R**)-**7** (102 mg, 92%). Colorless oil; ^1^H-NMR (400 MHz) δ = 0.57 (dd, *J* = 5.0 Hz, 5.0 Hz, 1H), 1.01 (dd, *J* = 5.0 Hz, 8.2 Hz, 1H), 1.19–1.27 (m, 2H), 1.31 (d, *J* = 6.0 Hz, 3H), 3.74 (d, *J* = 11.9 Hz, 1H), 3.91 (d, *J* = 11.9 Hz, 1H), 7.21–7.36 (m, 4H); ^13^C-NMR (125 MHz) δ = 13.9, 19.0, 19.2, 31.2, 66.7, 128.4, 130.3, 132.1, 143.1; ν_max_ (neat)/cm^−1^ 3402, 2930, 1493, 1036, 1016, 831. HRMS (DART) calcd for C_11_H_13_ClO [M + H]^+^ 196.0655, found 196.0651.

*(1R*,2R*)-1-(4-Chlorophenyl)-2-methylcyclopropane-1-methanol*: (1*R**,2*R**)-**7**



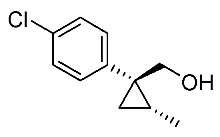



Following the similar reduction procedure described above, (1*R**,2*R**)-**7** was prepared from (1*R**,2*R**)-**6** (94%). Colorless oil; ^1^H-NMR (400 MHz, CDCl_3_) δ = 0.61 (dd, *J* = 4.8 Hz, *J* = 4.8 Hz, 1H), 0.80 (d, *J* = 6.2 Hz, 3H), 0.97 (dd, *J* = 4.8 Hz, *J* = 8.7 Hz, 1H), 1.06–1.14 (m, 1H), 1.25-1.45 (s, 1H), 3.38 (d, *J* = 11.2 Hz, 1H), 3.77 (d, *J* = 11.2 Hz, 1H), 7.24–7.32 (m, 4H); ^13^C-NMR (125 MHz) δ = 15.4, 16.1, 16.7, 32.8, 71.7, 128.3, 132.0, 132.2, 138.3; ν_max_ (neat)/cm^−1^ 3347, 2868, 1492, 1091, 1015, 827, 737.

*(1R,2S)-1-(4-Chlorophenyl)-2-methylcyclopropane-1-methanol*: (1*R*,2*S*)-**7**



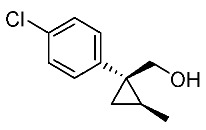



Following the similar reduction procedure described above, (1*R*,2*S*)-**7** was prepared from (1*R*,2*S*)-**6** (93%). Colorless oil; [α]${}_{D}^{19}$ +56.6 (*c* 1.12, CHCl_3_).

***(****1S,2R)-1-(4-Chlorophenyl)-2-methylcyclopropane-1-methanol*: (1*S*,2*R*)-**7**



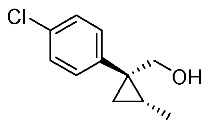



Following the similar reduction procedure described above, (1*S*,2*R*)-**7** was prepared from (1*S*,2*R*)-**6** (92%). Colorless oil; [α]${}_{D}^{19}$ −54.9 (*c* 0.46, CHCl_3_).

*(1R,2R)-1-(4-Chlorophenyl)-2-methylcyclopropane-1-methanol*: (1*R*,2*R*)-**7**



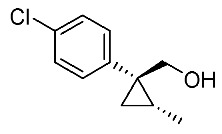



Following the similar reduction procedure described above, (1*R*,2*R*)-**7** was prepared from (1*R*,2*R*)-**6** (92%). White colored crystals; m.p. 56‒57 °C; [α]${}_{D}^{19}$ −31.9 (*c* 0.95, CHCl_3_).

*(1S,2S)-1-(4-Chlorophenyl)-2-methylcyclopropane-1-methanol*: (1*S*,2*S*)-**7**



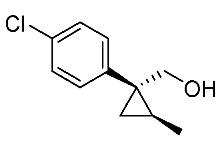



Following the similar reduction procedure described above, (1*S*,2*S*)-**7** was prepared from (1*S*,2*S*)-**6** (93%). White colored crystals; m.p. 51‒53 °C; [α]${}_{D}^{19}$ +32.4 (*c* 0.52, CHCl_3_).

*(1R*,2S*)-1-(4-Chlorophenyl)-2-methylcyclopropane-1-methyl 3-phenoxybenzyl ether*: (1*R**,2*S**)-**9**



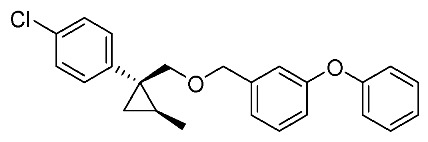



A mixture of (1*R**,2*S**)-**7** (183 mg, 0.93 mmol) and 3-phenoxybenzyl bromide (295 mg, 1.12 mmol) in DMF (0.9 mL) was added to a stirred suspension of NaH (60% dispersion; 45 mg, 1.12 mmol) at at 0–5 °C under an Ar atmosphere, and the mixture was stirred at 20–25 °C for 2 h. The mixture was quenched by water, which was extracted by EtOAc twice. The combined organic phase was washed with water, brine, dried (Na_2_SO_4_), and concentrated. The obtained crude oil was purified by SiO_2_-gel column chromatography (hexane/EtOAc = 30:1) to give the desired product (1*R**,2*S**)-**9** (322 mg, 91%). Colorless oil; ^1^H-NMR (400 MHz, CDCl_3_) δ = 0.53 (dd, *J* = 4.6 Hz, *J* = 4.6 Hz, 1H), 1.01 (dd, *J* = 4.6 Hz, *J* = 8.2 Hz, 1H), 1.14–1.28 (m, 4H), 3.51 (d, *J* = 10.1 Hz, 1H), 3.73 (d, *J* = 10.1 Hz, 1H), 4.39 (d, *J* = 12.4 Hz, 1H), 4.44 (d, *J* = 12.4 Hz, 1H), 6.90–7.01 (m, 5H), 7.10–7.27 (m, 6H), 7.32–7.36 (m, 2H); ^13^C-NMR (125 MHz) δ = 14.1, 19.5, 19.7, 28.8, 72.4, 73.9, 117.6, 117.8, 119.0, 122.1, 123.3, 128.1, 129.5, 129.7, 129.8, 131.5, 140.6, 143.9, 157.1, 157.4.; ν_max_ (neat)/cm^−1^ 2859, 1584, 1445, 1213, 1092, 1076, 957, 831, 748; HRMS (DART) calcd for C_24_H_23_ClO_2_ [M + H]^+^ 379.1465, found 379.1447.

*(1R*,2R*)-1-(4-Chlorophenyl)-2-methylcyclopropane-1-methyl 3-phenoxybenzyl ether*: (1*R**,2*R**)-**9**



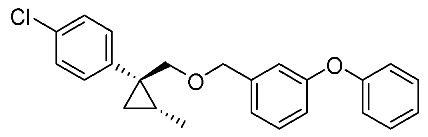



Following the similar etherification procedure described above, (1*R**,2*R**)-**9** was synthesized from (1*R**,2*R**)-**7** (79%). Colorless oil; ^1^H-NMR (400 MHz, CDCl_3_) δ = 0.62 (dd, *J* = 4.8 Hz, *J* = 4.8 Hz, 1H), 0.76 (d, *J* = 6.2 Hz, 3H), 0.94 (dd, *J* = 4.8 Hz, *J* = 8.7 Hz, 1H,), 1.03–1.11 (m, 1H), 3.33 (d, *J* = 9.9 Hz, 1H), 3.53 (d, *J* = 9.9 Hz, 1H), 4.38 (dd, *J* = 12.4 Hz, *J* = 14.7 Hz, 2H), 6.86–6.91 (m, 3H), 6.98–7.01 (m, 2H), 7.09–7.14 (m, 1H), 7.22–7.24 (m, 5H), 7.31–7.37 (m, 2H); ^13^C-NMR (125 MHz) δ = 15.4, 16.3, 17.0, 30.5, 72.1, 78.9, 117.4, 117.7, 118.9, 121.9, 123.2, 128.0, 129.5, 129.7, 131.9, 132.0, 138.9, 140.6, 157.0, 157.3; ν_max_ (neat)/cm^−1^ 2855, 1584, 1445, 1250, 1213, 1092, 1015, 743.

*(1R,2S)-1-(4-Chlorophenyl)-2-methylcyclopropane-1-methyl 3-phenoxybenzyl ether*: (1*R*,2*S*)-**9**



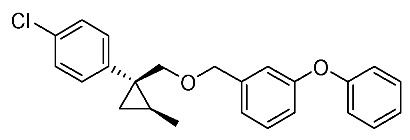



Following the similar etherification procedure described above, (1*R*,2*S*)-**9** was synthesized from (1*R*,2*S*)-**7** (85%). Colorless oil; [α]${}_{D}^{22}$ +25.8 (*c* 1.19, CHCl_3_).

*(1S,2R)-1-(4-Chlorophenyl)-2-methylcyclopropane-1-methyl 3-phenoxybenzyl ether*: (1*S*,2*R*)-**9**



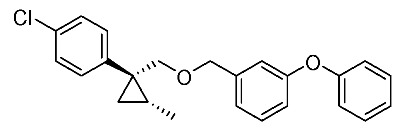



Following the similar etherification procedure described above, (1*S*,2*R*)-**9** was synthesized from (1*S*,2*R*)-**7** (72%). Colorless oil; [α]${}_{D}^{23}$ ‒24.5 (*c* 0.50, CHCl_3_).

*(1R,2R)-1-(4-Chlorophenyl)-2-methylcyclopropane-1-methyl 3-phenoxybenzyl ether*: (1*R*,2*R*)-**9**



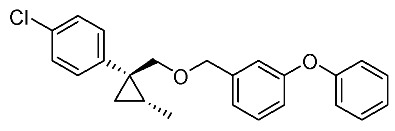



Following the similar etherification procedure described above, (1*R*,2*R*)-**9** was synthesized from (1*R*,2*R*)-**7** (91%). Colorless oil; [α]${}_{D}^{23}$ −31.8 (*c* 0.84, CHCl_3_).

*(1S,2S)-1-(4-Chlorophenyl)-2-methylcyclopropane-1-methyl 3-phenoxybenzyl ether*: (1*S*,2*S*)-**9**



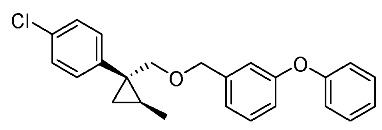



Following the similar etherification procedure described above, (1*S*,2*S*)-**9** was synthesized from (1*S*,2*S*)-**7** (97%). Colorless oil; [α]${}_{D}^{23}$ +32.4 (*c* 0.65, CHCl_3_).

*1-((((1R,2R)-1-(4-Ethoxyphenyl)-2-methylcyclopropyl)methoxy)methyl)-3-phenoxybenzene*: (1*R*,2*R*)-**10**



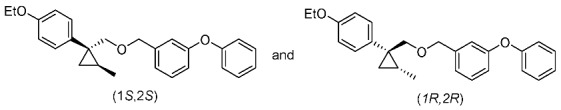



A mixture of (1*R*,2*R*)-**9** (108 mg, 0.29 mmol), Pd_2_(dba)_3_ (5.3 mg, 2 mol%), *t*Bu-XPhos (10 mg, 8 mol%) and KOH (81 mg, 5.0 eq.) in 1,4-dioxane (0.3 mL) and H_2_O (0.3 mL) was stirred at 100–105 °C for 16 h under an Ar atmosphere. The mixture was cooled down to room temperature, and quenched with water, which was extracted twice with EtOAc. The combined organic phase was washed with water, brine, dried (Na_2_SO_4_) and concentrated. The obtained crude oil was purified by SiO_2_-gel column chromatography (hexane/EtOAc = 20:1) to give the desired phenol product (95 mg, 90%).

K_2_CO_3_ (43 mg, 0.31 mmol) and EtI (122 mg, 0.78 mmol) was successively added to a stirred solution of phenol (92 mg, 0.26 mmol) in DMF (0.5 mL) at 0–5 °C under an Ar atmosphere. The mixture was stirred at 20–25 °C for 16 h. The mixture was quenched with water, which was extracted three times with EtOAc. The combined organic phase was washed with water, brine, dried (Na_2_SO_4_) and concentrated. The obtained crude oil was purified by SiO_2_-gel column chromatography (hexane/EtOAc = 20:1) to give the desired product (1*R*,2*R*)-**10** (84 mg, 83%).

(1*R*,2*R*)-**10**: Colorless oil; [α]${}_{D}^{21}$ −23.4 (*c* 0.95, CHCl_3_); ^1^H-NMR (400 MHz, CDCl_3_) δ = 0.60 (dd, *J* = 4.8 Hz, *J* = 4.8 Hz, 1H), 0.77 (d, *J* = 6.2 Hz, 3H), 0.90 (dd, *J* = 4.8 Hz, *J* = 8.2 Hz, 1H), 0.99–1.07 (m, 1H), 1.40 (t, *J* = 7.1 Hz, 3H), 3.34 (d, *J* = 9.9 Hz, 1H), 3.54 (d, *J* = 9.9 Hz, 1H), 4.00 (q, *J* = 7.1 Hz, 2H), 4.39 (t, *J* = 13.3 Hz, 2H), 6.78–6.82 (m, 2H), 6.86–6.88 (m, 2H), 6.93 (d, *J* = 8.0 Hz, 1H), 6.98–7.00 (m, 2H), 7.08–7.12 (m, 1H), 7.19–7.22 (m, 2H), 7.23–7.25 (m, 1H), 7.31–7.35 (m, 2H); ^13^C-NMR (125 MHz) δ = 14.9, 15.4, 16.2, 17.0, 30.2, 63.2, 72.1, 79.3, 113.9, 117.6, 117.7, 118.9, 122.0, 123.2, 129.5, 129.7, 131.5, 132.2, 140.9, 157.2, 157.2, 157.4; ν_max_ (neat)/cm^−1^ 2980, 2868, 1611, 1584, 1514, 1487, 1445, 1393, 1354, 1288, 1242, 1215, 1175, 1140, 1107, 1074, 1049, 959, 924, 835, 789, 735, 692; HRMS (ESI): *m*/*z* calcd for C_26_H_28_O_3_ [M + Na]^+^ 411.1936; found: 411.1936.

(1*R*,2*R*)-**10**: Pale yellow oil; [α]${}_{D}^{23}$ +24.7 (*c* 0.21, CHCl_3_).

## 4. Conclusions

We envisaged syntheses of simple and accessible novel cyclopropane-type pyrethroids with two chiral centers on the cyclopropane ring: cyanohydrin ester-type **1** and ether-type **2**. The design of **1** and **2** involved a “reverse connection approach” derived from fenvalerate and etofenprox, respectively. The synthesis of chiral 2-methylcyclopropanes **1** and **2** were performed by accessible ring-opening reactions of 4-chlorobenzyl cyanide with commercially available (±)-, (*R*)-, and (*S*)-propylene oxides as the crucial step, and provided the products in good overall yield with every >98% ee. During the synthetic study, we also performed alternative convergent syntheses of etofenprox and 4-EtO-type cyclopropane analogues from the parent 4-*Cl*-type pyrethroids utilizing a hydroxylation cross-coupling reaction that was recently developed by Stradiotto’s group.

The bioassay using common mosquito revealed that none of the four cyanohydrin ester-type **1** compounds exhibited insecticidal activity, whereas among the four ether-type compounds **2**, only the (1*R*,2*R*)-**2** enantiomer had significantly more activity than (1*S*,2*S*)-**2**. The present clear chiral discrimination was also observed for the previously reported “cyanohydrin ester-type cyclopropane” pyrethroids with three chiral centers. The subtle nature of the stereostructure-activity relationships is likely a privileged trend in pyrethroid chemistry, and the present finding will contribute to future discovery in this field.

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
