# Peer review of "Synthesis and Stereostructure-Activity Relationship of Novel Pyrethroids Possessing Two Asymmetric Centers on a Cyclopropane Ring"

_molecules, 2019, doi:10.3390/molecules24061023_

Round 1

Reviewer 1 Report

In this manuscript, Tanabe describes two strategies for the synthesis of pyrethroids. Application of these strategies for the synthesis of etofenprox and some analogues is also reported. Insecticidal activity of some of the synthesized compounds is assessed allowing to investigate the structure-activity relationship of these pyrethroid derivatives.

The first strategy allows controlling the absolute stereochemistry of the quaternary stereogenic center of the cyclopropane of pyrehroids, by starting from enantiopur epoxide (see Scheme 6). A weakness of this work is however that none of the reported synthesis are diastereoselective.

The originality of the two strategies and the interest of this work for the chemistry community are however sufficient in my opinion to justify its publication in Molecules.

Remarks:

- Page 3, below scheme 2, the authors mention the work of Hiyama and Takehara. A reference should be provided.

- Table 1, footnote a mention that reported yield were determined by 1H NMR on the crude mixture. How these yield were determined, did the authors used an internal reference?

- Page 8, General section of Materials and Methods: the authors write that 13C NMR were recorded  at 100 and 120 MHz. "120" should be "125".

p { margin-bottom: 0.1in; line-height: 120%; }

Author Response

> Page 3, ….

A reference [16] was provided.

> Table 1, ….

“IS” is ethylene carbonate.  This was commented.

> Page 8, ….

Yes, I altered “120 MHz” to “125 MHz”.

Reviewer 2 Report

Manuscript by Tanabe et al. describe synthesis and biotesting of new compounds which can act as potential pyrethroid insecticides. The new structures with cyclopropane ring as a key structural motif have been designed based on compounds with well-known pyrethroid activity, such as Etofenprox and Esfenvalarate. Due to presence of two stereocenters in the cyclopropane ring, four distinct stereoisomers can exist for the each of the target compounds, which can exhibit different bioactivity. Therefore, synthesis of all these stereoisomers might be required. Preliminary screening of bioactivity has been performed for racemic compounds, followed by synthesis of all stereoisomers for the promising candidates. It has been found that cyclopropane analog of Etofenprox is active against the common mosquito, and chiral discrimination between (1S,2S)- and (1R,2R)-isomers has been observed. Etofenprox and Fenvalarate have been synthesized and used as control samples. In general, the research has been perfectly planned and carefully performed. Experimental part contains enough detail and all new compounds are fully characterized. Perhaps, the discussion could become even more stronger if the authors could provide a rationale for distinct bioactivity observed for different stereisomers of Etofenprox analogues, and its absence for the synthetized Fenvalarate analogues, e.g. based on computation docking studies with model of pyrethroid receptor.

The authors present synthesis of new pyrethroids and their stereostructure-activity relationship. The manuscript can be published in Molecules after minor corrections, see below.

1) lines 63-67: "Ring-opening of 4-chlorobenzyl cyanide with...": chlorobenzyl cyanide does not contain any ring to cleave. Please reformulate. Overall, this sentence is too long and hard to follow. Suggest rewriting.

2) Scheme 2 caption: please remove additional period at the end

3) line 76: "Hiyama and Takehara's group reported..." - please place appropriate citation here

4) line 82: "at a higher position": please use "higher chemical shift" or similar.

5) Scheme 4 caption: please remove additional period at the end

6) line 99: please correct reference format (9 is superscripted)

7) Scheme 5 caption: please remove additional period at the end

8) Scheme 7: considering transformation of Cl into OEt, it would be better to place the structure of the phenol intermediate (in square brackets). It would help the reader to understand this transformation better. In subsequent schemes it could be skipped.

Author Response

> 1)

Yes, I rewrite the sentence, easier to read.

> 2) , > 5, > 7

I removed additional periods at the end.

> 3)

Yes, I provided the reference [16].   

> 4)

Yes, I altered “position” to “chemical shift.”

> 6)

I altered it.

> 8)

Yes, following your reasonable suggestion, I inserted the phenol intermediate in a square bracket.

Reviewer 3 Report

This manuscript merits publication in Molecules, because it reports interesting chemistry and biological results., but major changes are required.   1.  The English wording is confuse and poor. It mus be substantitally improved. Revison by a native English spoken person is recommendable.   2.  The first paragraph of the Abstract section is confuse and wrong: the two types of  2-methylcyclopropane pyrethroid insecticides reported should be named more precisely. Both of them have three stereogenic centers (not two). The concept "reverse connection approach" is unknown in the chemistry literature.The second paragraph is too long and sometines confuse. It should be improved.   3.  Chemical concepts are sometime used in an improperly way along de body of the manuscript.   4.  Scheme 1 is wrong. It's is not a retrosynthetic scheme. Retrosynthetic arrows are used improperly. I suggest the authors to join Figure 1, Figure 2 and Figure 3 in an only figure as follows: a) row 1 should include Esfenvalerate, Etofenprox and Cyclopothrin (please, specify X,X). b) Second row: should include compounds I, II and III. c) Third row should include compound 1 and 2.   5.  Scheme 2 states a synthetic plan for the racemic (row 1) and stereoselective (rows 2 and 3)  preparation of 1-(p-chlorophenil)-2-methylcycloprane nitriles, but the racemic preparation was the only studied. The stereoselective version of this reaction is much more interesting from the chemical point of view. So the author should be asked to study them.

Author Response

> 1

I would like to be checked by MDPI editing service.

> 2

The present 2-methylcyclopropane pyrethroids have clearly two chiral centers.  "reverse connection approach" is an our design concept.  I hope the phrase as it stands.

I altered “4-chlorobenzyl cyanide” to “4-chlorobenzyl cyanide anion”.  The sentence is not so long.

> 3

I cannot understand the opinion.

> 4

I changed retrosynthetic arrows to dotted arrows.

I cannot accept your suggestions.  Please be as it stands.

> 5

No opinion. 

Round 2

Reviewer 3 Report

This manuscript merits publication in Molecules, because it reports interesting chemistry and biological results. 

The answers by the authors to my previous comments are partially satisfactory, as it can be seen from my present following comments:

1.     Author note: I would like to be checked by MDPI editing service.

Reviewer comment: OK

2.     Author note: The present 2-methylcyclopropane pyrethroids have clearly two chiral centers. 

Reviewer comment: Pyrethroids 2 have certainly two stereogenic (chiral) centers, but pyrethroids 1 have three: two on the cyclopropane ring and one at the carbon bearing the CN group.

3.     Author note: "reverse connection approach" is an our design concept.  I hope the phrase as it stands. 

 Reviewer comment: OK

4.     Author note: I altered “4-chlorobenzyl cyanide” to “4-chlorobenzyl cyanide anion”.  The sentence is not so long. 

Reviewer comment: OK

5.     Author note: I cannot understand the opinion 

 Reviewer comment: Please, pay close attention to the changes and suggestions stated in the revised manuscript (see the attached pdf document).

6.     Author note: I changed retrosynthetic arrows to dotted arrows. . I cannot accept your suggestions.  Please be as it stands.. 

Reviewer comment: OK

7.     Author note: No opinion

Reviewer comment: OK

Assuming that the MDPI services will revise the English wording, I consider that the article can be published in Molecules after acceptance or justified rejection of the changes stated in the attached pdf document..

Author Response

Concerning the suggestion of p. 5, I hope for submission of the contents, as it stands.  The present chiral synthetic part really appears in this section, with maintaining the adequate consistency.    
